



# 3D geological modelling of igneous intrusions in LoopStructural v1.5.10

Fernanda Alvarado-Neves[1], Laurent Ailleres[1], Lachlan Grose[1], Alexander Cruden[1], Robin Armit[1]

[1] School of Earth, Atmosphere and Environment, Monash University PO Box 28E, Victoria, Australia

*Correspondence to*: Fernanda Alvarado-Neves (Fernanda.alvaradoneves@monash.edu)

**Abstract.** Over the last two decades, there have been significant advances to improve the 3D modelling of geological structures by incorporating geological knowledge into the model algorithms. These methods take advantage of different structural data types and do not require manual processing, making them robust and objective. Igneous intrusions have received little attention
in 3D modelling workflows, and there is no current method that ensures the reproduction of intrusion shapes comparable to those mapped in the field or in geophysical imagery. Intrusions are usually partly or totally covered, making the generation of realistic 3D models challenging without the modeller's intervention. In this contribution, we present a method to 3D model igneous intrusions considering geometric constraints consistent with emplacement mechanisms. Contact data and inflation and propagation direction are used to constrain the geometry of the intrusion. Conceptual models of the intrusion contact are fitted
to the data, providing a characterisation of the intrusion thickness and width. The method is tested in synthetic and real-world case studies and the results indicate that the method can reproduce expected geometries without manual processing and with restricted datasets. A comparison with Radial Basis Function (RBF) interpolation shows that our method can better reproduce complex geometries such as saucer-shaped sill complexes.

## 1 Introduction

Significant advances in 3D geological modelling have shown that incorporating prior geological knowledge into interpolation algorithms can significantly improve the 3D representation of the geometry of structures (e.g., Godefroy *et al.*, 2017; Grose *et al.*, 2018; Grose *et al.*, 2019; Hillier *et al.*, 2014; Laurent *et al.*, 2013, 2016; Thibert *et al.*, 2005). Geological knowledge of a geological feature can be incorporated into the 3D modelling workflow using different approaches. For instance, by parameterising its 3D geometry, defining its expected geometries, or using complete structural datasets. These approaches
have been applied to folds and faults, showing substantial improvements in 3D geological models, especially in models built using few or poor-quality observations.

In the case of igneous intrusions, there are no methods that incorporate prior knowledge into the modelling algorithm. Implicit 3D models of intrusions are currently characterised by a surface representing its contact boundary. This boundary is
numerically described using the same frameworks as those used to build other geological interfaces, such as stratigraphic contacts or faults (Wellmann and Caumon, 2018, Calcagno *et al.*, 2008). However, intrusions' geometry differs from these



geological features because they are closed surfaces that are not continuous in the 3D space. Two types of constraints are generally applied to build 3D models of igneous intrusions: point data that indicate the location of the contact between the intrusion and the host rock, and a polarity constraint which is a vector indicating the direction from the outside to the inside of

the intrusion (Calcagno *et al.*, 2008). The distinction between the top and base contact of the intrusion and other field measurements, such as the inflation and flow direction of the magma, are not considered to constrain the models. While the polarity constraint helps to adapt current interpolation methods to intrusions, it is not measurable in the field and does not have any geological meaning.

Igneous intrusions, such as plutons, laccoliths, sills and layered intrusions, develop tabular bodies, with their horizontal dimension greater than their vertical dimension (e.g.*,* Cruden *et al.*, 2017; Cruden *et al.*, 1999; McCaffrey and Petford 1997; Vigneresse *et al.*, 1999). Their geometries and locations in the crust are strongly controlled by the anisotropies of the host rock that facilitated their emplacement, such as beddings and faults (e.g., Barnett and Gudmundsson 2014; Brun and Pons 1981; Clemens and Mawer 1992; Gudmundsson 2011; Morgan 2018; Souche *et al.*, 2019). While intrusions 3D models estimated

with existing methods are consistent with contact observations, they may not honour the tabular nature of intrusions without manual processing in sparse data environments. In particular, the intrusion shape and its geometrical relation with the host rock are unlikely to be captured away from the data. This is particularly important for 3D models of intrusions since they are usually only partly exposed if not totally covered and consequently inferred from geophysical interpretations or simulations, and intrusion observations (location and orientation of the contacts) are usually sparse.


To address the problem of poor 3D representation of intrusions, we propose a general workflow inspired by the Object-Distance Simulation Method (ODSIM, Henrion *et al.,* 2008; 2010). Our method integrates conceptual knowledge of magma emplacement mechanisms into the ODSIM framework, enabling the reproduction of intrusion geometries comparable to those observed in reality. As these concepts are integrated into the method framework, the results are objective and reproducible. In

practice, the method can use different types of datasets, build models of different types of intrusions, and incorporate knowledge of magma's mechanical behaviour into a purely geometric approach. The approach has three main steps. We initially build a structural frame adapted for intrusions (Grose *et al.*, 2021a, b). This object is a curvilinear coordinate system whose main axis represents the location of the intrusion's top (or base) contact. The intrusion frame is constrained using contact data, the geometry of the host rock's foliation and/or structures that facilitated the emplacement of the intrusion, and vector

directions indicating the propagation and growth of the magma. Then, conceptual models describing the coarse-scale geometry of the intrusion are parametrised using the intrusion frame coordinates and are employed to characterise the contact geometry along its axes. Finally, we use the conceptual models to modify the intrusion frame scalar fields to obtain a unique scalar field whose isovalue 0 represents the intrusions' contact boundary.



This contribution is organised as follows. First, we summarise intrusion emplacement mechanisms and geometries of intrusions described in the literature. Secondly, we introduce the method developed in this work and its association with previous work. Thirdly, we show the results of the application of the method in three case studies: a laccolith, and pluton, and a sill complex. Then, we assess the value of this method by comparing the resulting 3D model of a sill intrusion with its 3D model built using a classical interpolation framework. Finally, we discuss the advantages of adding geological knowledge of intrusions in the

modelling framework, the limitations of our method, and further work that can be done to improve this approach.

## 2 Igneous intrusions: general overview

Igneous intrusions comprise a significant volume of the Earth's crust and are found in all tectonic settings. They are part of Volcanic and Igneous Plumbing Systems, which involve magma production, transport and emplacement (Burchardt 2018). Magma production occurs due to partial melting of rocks in the upper mantle or crust (e.g., Brown 2007; Petford *et al., 2*000;

van Wyk de Vries and van Wyk de Vries 2018). Magma can be vertically and laterally transported to its final emplacement location by the intrusion of dykes, sills and inclined sheets (e.g., Brown 2007; Magee *et al., 2*016). The emplacement of magma is controlled by mechanical interactions and the density contrast between the magma and its surroundings (e.g., Brown 2007; Hutton 1988a; Petford *et al., 2*000).

The emplacement of the magma is initiated when a vertically propagating magma conduit (*i.e.*, dyke) is arrested. Regardless of the magma composition and depth of emplacement, host rock heterogeneities and mechanical properties strongly control the intrusion location and final morphology. Examples of these are a stiffness contrasts between adjacent layers (e.g., Barnett and Gudmundsson 2014; Brun and Pons 1981), unconformities (e.g., Hogan and Gilbert 1995), host rock discontinuities (e.g., Clemens and Mawer 1992), stress barriers (e.g., Barnett and Gudmundsson 2014), and shear zones (e.g., Guineberteau *et al.,*

*1*987; Weinberg *et al., 2*004).

Once magma has been arrested, intrusion growth is controlled by host rock anisotropies until it reaches its maximum lateral and vertical extent. The growth of plutons depends on host rock mechanical properties (Cruden and Weinberg, 2018) and can occur by both vertical and/or lateral displacement of the host rock (e.g., Cruden 1998; Cruden *et al., 1*999; Grocott *et al.,*

*1*999). Sills grow by horizontal propagation of their lateral tips and by vertical inflation (e.g., Hutton 2009). If two or more sill segments propagate in the same direction, but at different stratigraphic levels, they eventually coalesce, developing connectors such as steps or bridges (e.g., Hutton 2009; Köpping *et al.*, 2022; Magee *et al.*, 2019; Schofield *et al.*, 2012). The sill inflation direction is parallel to the intrusion opening vector, which may or may not be orthogonal to the intrusion plane (Magee *et al.*, 2019). Laccoliths are generally developed by the incremental growth of an initial sill (e.g., Annen *et al.*, 2015; Chen and

Nabelek 2017; Johnson and Pollard 1973; Michel *et al.*, 2008). The sills' top and bottom contacts act as discontinuities controlling the emplacement of new sheets (Morgan 2018).





The geometries of intrusions have been characterised using different datasets, such as field observations of the intrusion's top, base and lateral contacts, drilling data, and interpretation of gravity and seismic surveys (e.g., Braga *et al.,* 2019; Cervantes 2019; Eshaghi *et al.,* 2016; Rawling *et al.,* 2011, Groccot *et al.,* 2009, Leaman *et al.,* 1976, 2002, Paterson *et al.,* 1996). There is a general agreement that intrusions develop tabular geometries in the coarse scale with their horizontal dimensions greater than their vertical dimension (e.g., Cruden *et al.,* 2017; Cruden *et al.* .1999; McCaffrey and Petford 1997; Vigneresse *et al.* .1999). On the smaller scale, intrusions show a variety of shapes (Figure 1). Plutons can be symmetric or asymmetric with one or more vertical feeder zones (Clemens and Mawer 1992; Vigneresse 1995; Vigneresse *et al.,* 1999). In a plan view, plutons show elliptical or irregular geometry (Cruden 1998). Their roof is roughly planar with an abrupt roof-sides transition (Patterson *et al.,* 1996), and the floor may be wedge-shaped, dipping towards the feeder, or tablet-shaped concordant with the roof (Cruden and McCaffrey 2001; Cruden 2006; Vigneresse 1995; Vigneresse *et al.,* 1999). Sills are sheet-like intrusions that can develop strata-concordant tabular bodies with little or no change in thickness or straight or step-wise transgressive bodies developing an oblique angle with the host rock foliation (Galland *et al.,* 2018; Jackson *et al.,* 2013). Sills may also develop saucer or V-shapes, with a thicker concordant inner sill that transitions to thinner transgressive outer sills (Galland *et al.,* 2018; Köpping *et al.,* 2022). Sill networks are composed of sill segments, which are generally elongated and narrow in map view, with tablet-shape or elliptical cross-sections (Leaman, 1995; Schofield *et al.,* 2010, 2012). Laccolith roof and sides may be symmetric, developing a bell-jar shape with a slightly arched and concordant roof and outward dipping sides (e.g., Clemens and Mawer 1992; Johnson and Pollard 1973; Morgan 2018), or asymmetric with a flat roof concordant to the host rock layering and bounded by a fault on one side (e.g., de Saint-Blanquat *et al.* 2006). The floor of laccoliths is usually concordant to the stratigraphy with one feeder zone.

**Figure 1. Schematic intrusion shapes and field examples. (a) Plutons: schematic cross-section of plutons' roof (after Paterson _et al._, 1996) and tablet-shaped floor contact inferred from 3D inversion of gravity data (after Vigneresse _et al._, 1999). Field example**
**showing the roof of San Gabriel pluton emplaced in the volcano-sedimentary Abanico Formation, Maipo Valley, Central Chile. (b) Schematic morphologies of sill sheets (after Galland _et al._, 2018) and a field example from the roof contact of the Tasmanian Dolerite emplaced in the sedimentary Parmeener Supergroup. (c) Schematic map view and cross-section of sill complex developing bridges and sill connectors (_a.k.a._ broken bridges) from Keopping _et al._ (2021). Field examples from Theron Mountains (Hutton 2009). (d) Schematic cross-section of roof and floor of laccoliths (after Johnson and Pollard, 1973) and photograph of the roof and floor contact**
**of Torres del Paine laccolith, Patagonia, Chile.**





## 3 Three-dimensional modelling of intrusions using constraints from emplacement mechanisms

In this contribution, we present a method to build implicit geological models of intrusions that integrates current knowledge on emplacement mechanisms and that honour intrusions geometries described in the literature (see Section 2). This is achieved by:


- building an *intrusion frame*, a local coordinate system that represents the main geometrical elements of the intrusion,
- parametrising *conceptual models* using the intrusion frame coordinates to estimate the intrusion lateral and vertical contact, and
- computing an implicit representation of the intrusion using the conceptual models and the intrusion frame scalar fields.


The method is implemented in the *intrusion* module of *LoopStructural* (Grose *et al.*, 2021a, b), and an intrusion can be built using the *create_and_add_intrusion* function from the GeologicalModel application programming interface.

### 3.1 Method overview

Our method is inspired by the Object-distance Simulation Method proposed by Henrion *et al.* (2008, 2010). The Object-
distance simulation method (ODSIM) was developed to model geological bodies whose geometry is affected by pre-existing geological features, such as karsts. The ODSIM models a three-dimensional scalar field around a skeleton. The skeleton object can be constructed deterministically or by using object-based or stochastic simulations. A distance scalar field is computed around the skeleton and is perturbed using a stochastically generated random threshold. The geological body is defined using an indicator function as follows:


$$I_B(p) = \begin{cases} 1 & if\ D(p) \leq \varphi(p) \\ 0 & otherwise \end{cases} \tag{1}$$

Where $D(p)$ is the distance scalar field computed around the skeleton over the points of a previously defined grid $G$, and $\varphi(p)$ is the random threshold. The data conditioning is obtained when $\varphi(p)$ is higher than or equal to the distance between the data and the skeleton. It can be reached by transforming the data to threshold values after the simulation of $\varphi(p)$, or by using the
data to condition $\varphi(p)$ if using Sequential Gaussian Simulation (e.g., Clausolles *et al., 2019*).

For modelling igneous intrusions, we replace the skeleton and the distance scalar field with a structural frame (Grose *et al.*, 2021 a, b). A structural frame is a curvilinear coordinate system composed of three axes, each representing a major structural direction of the modelled geological feature and bearing a scalar field implicitly defined throughout the model. The method
does not use a skeleton per se because intrusions are frequently not entirely exposed, and only roof or floor contacts can be





mapped, making it challenging to identify the centre line of a body. Also, the structural frame allows us to integrate conceptual knowledge of emplacement mechanisms into the algorithm. Existing implementation of fold and faults structural frames in LoopStructural allows to parameterise the folded and faulted foliations at any point in the model, enabling the reproduction of highly deformed terrains (Grose *et al.*, 2021a, b).


We use geometrical conceptual models of the intrusion geometry to modify the scalar fields. The conceptual models are essentially parametric functions that describe the intrusion's coarse scale geometry and allow integration of the interpreted intrusion shapes into the method algorithm. The functions are parameterised using the coordinates of the structural frame and are afterwards fitted to the observations of the intrusion contact. The fitted conceptual models represent distance thresholds

characterising the intrusion contact along the structural frame coordinate.

To obtain an implicit representation of the intrusion, we modify the intrusion frame scalar fields using the distance thresholds given by the conceptual models. We combine the intrusion frame scalar fields into one scalar field, whose isosurface 0 represents the intrusion's contact.




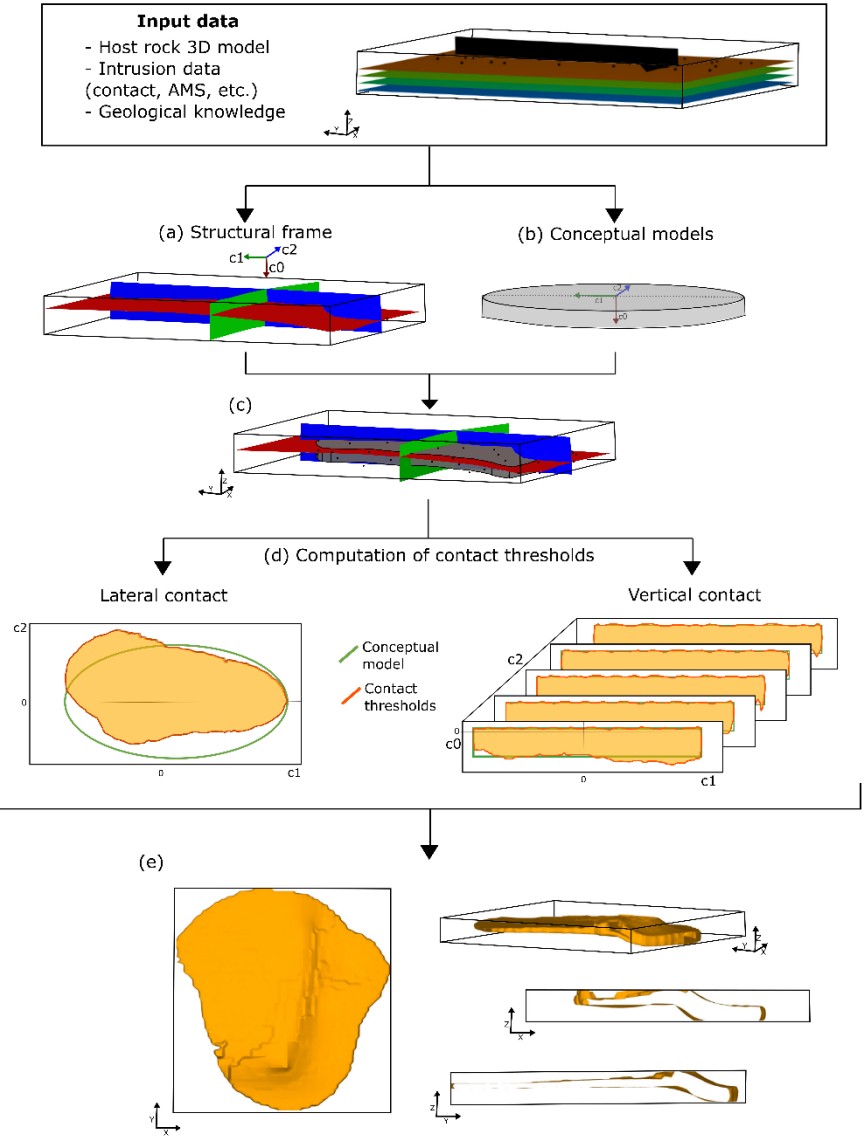

**Figure 2. Workflow for the proposed method using a synthetic example of a sill intrusion. The data and prior knowledge indicate that the sill exploits the host rock's bedding and two faults to step up in the stratigraphy. (a) The structural frame's coordinates are built using the following data: the geometry of the bedding and faults for coordinate 0, propagation data for coordinate 1, and synthetic vectors perpendicular to the sill's long axis for coordinate 2. (b) The conceptual models used are the ellipsis and a constant function to constrain the lateral and vertical extent, respectively. (c) Conceptual models and structural frame observed along XYZ axes. (d) Conceptual models (green lines) and conditioned conceptual models (orange lines) observed along the structural frame axes. (e) Different views of the isosurface that represents the intrusion contact.**





## 3.2 Intrusion structural frame

The intrusion frame is built using LoopStructural implementation of structural frames (Grose *et al.*, 2021a, b), in which the coordinates are interpolated sequentially using a discrete interpolator, e.g.*,* finite difference interpolator on a cartesian grid (Irakarama *et al.*, 2020) or piecewise linear interpolation on a tetrahedral mesh (Frank *et al.*, 2005, 2007).

The intrusion frame coordinates represent geometrical elements of the intrusion shape (Figure 2a). The first coordinate ($c_0$)
measures the distance to the roof or floor contact of the intrusion. Its scalar field is interpolated using contact observations and is constrained to be parallel to the foliation and/or structures that facilitated the emplacement of the intrusion. The gradient of this coordinate is forced to be perpendicular to the host rock's anisotropies, unless inflation measurements are available. The isosurface $c_0=0$ approximates the location of the roof or floor contact (depending on the data used to constrain this coordinate). The second coordinate ($c_1$) describes the propagation of the magma, and is interpolated using measurements (or geological
knowledge) of the propagation direction. Conceptually, the isovalue $c_1=0$ should be related to the position of the intrusion feeder. However, for the modelling, this isosurface can be anywhere in the model. The third coordinate ($c_2$) measures the distance to the long axis of the intrusion. It is interpolated using points along the intrusion long axis and an additional constraint enforcing the orthogonality between the gradients of $c_1$ and $c_2$.

## 3.3 Conceptual models and threshold functions

The intrusion frame coordinates are used to parameterise two conceptual models that represent the coarse scale geometry of the intrusion (Figure 2b). These conceptual models are simple geometric shapes observed along the frame coordinates, however they may show a more complex geometry observed within the *xyz* coordinate system (Figure 2c). The first conceptual model, $\acute{C}_L(c_1)=c_2$, returns a distance along $c_2$ for any $c_1$, and represents the geometry of the intrusion lateral contact. The second conceptual model, $\acute{C}_V(c_1,c_2)=c_0$, returns a distance along $c_0$ for any ($c_1,c_2$), and represents the geometry of the roof or floor
contact of the intrusion, depending on which of these contacts were used to build the intrusion frame. For example, if the intrusion frame $c_0$ is built using roof contact points, $\acute{C}_V$ represents the geometry of the intrusion floor.

To fit these models to the contact data, we first interpolate the residual values between the data and the conceptual model at the data locations (Figure 2d). Then, we define threshold distances functions $T(p)$ as the difference between the conceptual
model and the interpolation. This allows us to honour both the data and the interpreted intrusion geometry and create models with small datasets.

Consider a set of roof or floor contact points $p^i$ with $i=\{0,…, n\}$, and a set of lateral contact points $p^k$ with $k=\{0,…, m\}$ and their associated intrusion frame coordinates ($c_0^i,c_1^i,c_2^i$) and ($c_0^k,c_1^k,c_2^k$), respectively. We define the residual values $R_V$ and $R_L$
as:


$$R_V(c_1{}^i, c_2{}^i) = \acute{C}_V(c_1{}^i, c_2{}^i) - c_0{}^i \quad ; \quad \forall\, i = \{0, \ldots, n\}$$

$$R_L(c_1{}^k) = \acute{C}_L(c_1{}^k) - c_2{}^k \quad ; \quad \forall\, k = \{0, \ldots, m\}$$


Where $\acute{C}_{V\text{ and }}\acute{C}_L$ are the geometrical conceptual model for the vertical and lateral contact, respectively. We interpolate between the residual values using an exact interpolator, and we obtain the interpolators $\check{R}_V(c_1, c_2)$ and $\check{R}_L(c_1)$. Finally, we define the distance threshold functions $T(p)$ as:


$$T_V(c_1, c_2) = \acute{C}_V(c_1, c_2) - \check{R}_V(c_1, c_2)$$

$$T_L(c_1) = \acute{C}_L(c_1) - \check{R}_L(c_1)$$

Interpolating the residual values using an exact interpolator allow us to condition the model to the data at this step. However, other interpolation techniques can be used and condition the model to the data afterwards.


Using the threshold functions along the intrusion frame coordinates, the intrusion body $I$ can be defined as:

$$I = \{(c_0, c_1, c_2) \mid 0 \leq c_0 \leq T_V(c_1, c_2) \text{ and } c_2 \leq T_L(c_1)\}$$

### 3.4 Implicit description of the intrusion geometry

The implicit description of the intrusion can be obtained by modifying the intrusion frame scalar fields, so the intrusion contact characterised by the threshold functions along the frame coordinates is represented by the isosurface 0 of this modified scalar field (Figure 2e). This can be achieved by different combinations of the scalar fields and threshold functions.

### 4 Results

In this section, we present three case studies that show the applications of our approach to different types of intrusions: a sill
complex, a pluton and a laccolith. These examples are presented as interactive *jupyter* notebooks that can be downloaded from *https://doi.org/10.5281/zenodo.8189191*. We also present a comparison between our method and Radial Basis Functions interpolation using an example of a sill complex offshore of Western Australia.

### 4.1 Case Study 1: Synthetic sill complex

The first case study (CS1) is a synthetic sill complex composed of three sill segments. The sill complex is emplaced in a
horizontal stratigraphic sequence, and the sill segments propagate to the north, with slightly different directions. Two of the





sill segments (segments 0 and 1) were intruded at the same stratigraphic level, while the middle segment (segment 2) exploited a pre-existing EW trending fault and stepped up in the stratigraphy.

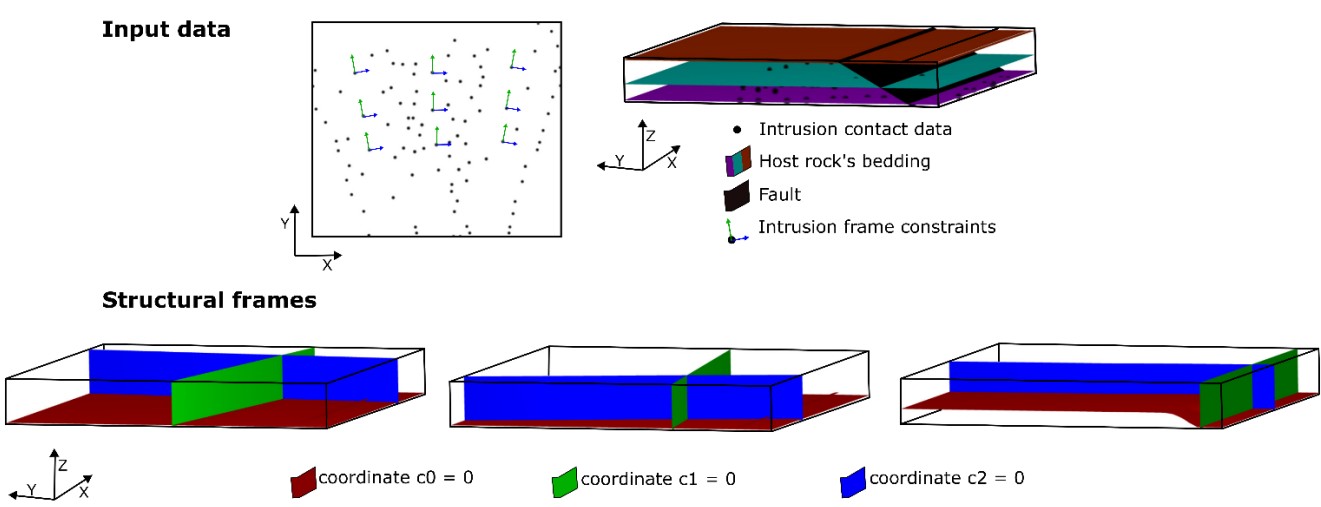

**Figure 3. Input data and structural frames of Case Study 1 - Synthetic sill complex. The dataset consists of the 3D model of the host rock, roof and floor contact points, propagation data and synthetic vectors perpendicular to the long axis of each sill.**

In this example, the input data consisted of an implicit geological model of the stratigraphic sequence and the fault, contact data of the roof, floor and sides of each sill segment, and propagation vectors and points located at the long axis of each segment. Figure 3 shows the 3D geological model of the host rock and the spatial distribution of the sill segments' data. The intrusion frame of each segment is built using the floor contact point and propagation and long axis data (Figure 3). The three sill segments are built with the same conceptual models: the ellipsis equation as the lateral contact conceptual model $\dot{C}_L$ and a constant function as the vertical conceptual model $\dot{C}_V$.

$$\dot{C}_L : \frac{(c_1 - c_1')}{a^2} - \frac{(c_2 - c_2')}{b^2} = 1$$

$$\dot{C}_V : c_0 = c_0^{mean}$$

Where $(c_0', c_2')$ is a point chosen arbitrarily in the centre of the intrusion considering the data spatial distribution, $a, b$ and $c$ are the average of the $c_1, c_2$, and $c_0$ coordinate values of the input data, respectively. Considering that $c_0=0$ approximates the location of the floor, $\dot{C}_V$ is equivalent to the mean thickness of each sill. Figure 4 shows the 3D geological model of this case





study. Section views along the Y axis show structures usually developed in sill complexes, like broken bridges when sills inflate and coalesce or bridges when they inflate without coalescing.

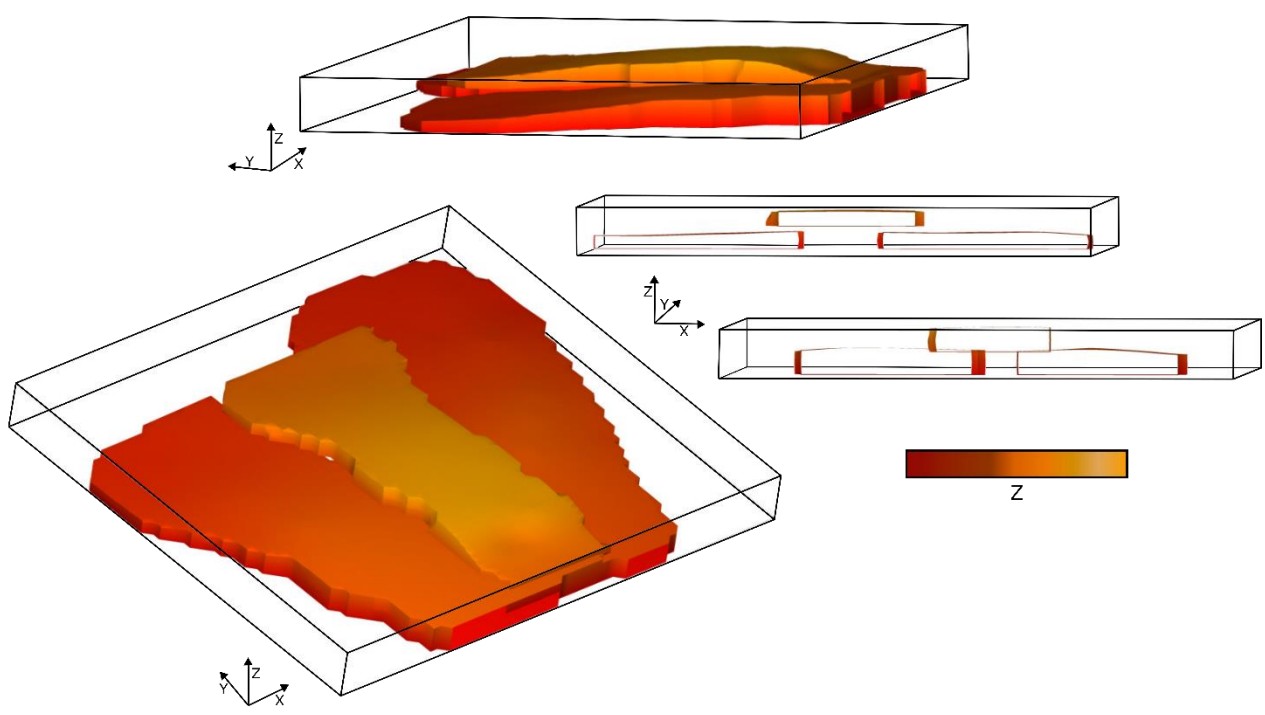

**Figure 4. 3D geological models of Case Study 1 – Synthetic sill complex. To the right, two cross-sections show bridge and broken bridge structures developed between the sills. The isosurfaces are painted with the elevation value at each location, highlighting the relief of the models.**

## 4.2 Case Study 2: Voisey's Bay intrusion

The second case study is the Voisey's Bay intrusion in Labrador, Canada. In this case study, we created the dataset by selecting intrusion contact data points from the geological map and geological cross-sections presented by Saumur and Cruden (2015). The floor data points were picked from the drill holes in the interpreted cross-sections. The roof and lateral data were picked from the geological map; therefore, it is assumed that the roof is located in the current topography. The host rock was modelled as a horizontally foliated unit. Figure 5 shows the 3D model of the host rock and the contact data points.

Considering the spatial distribution of the contact data, we approximate the long axis of the intrusion as a SE-NW line centred in the intrusion. The intrusion frame coordinate $c_0$ is constrained using the roof contact data. Coordinate $c_1$ is constrained to be parallel to the long axis, and coordinate $c_2$ perpendicular to the long axis. Figure 5 shows the intrusion frame of this case study.



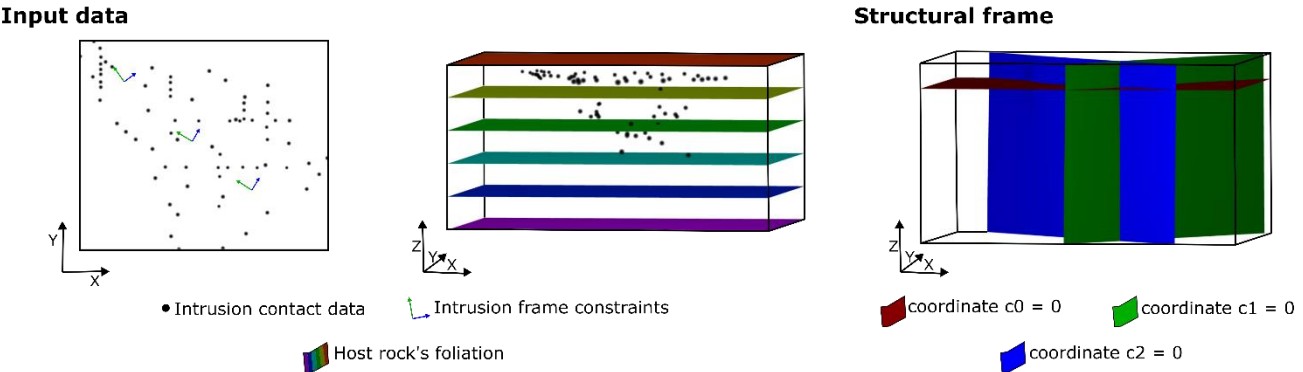

**Figure 5. Input data and structural frames of Case Study 2 – Voisey's Bay intrusion, Canada. The dataset consists of a 3D model of the host rock and roof and floor contact points extracted from the area's geological maps and cross-sections of the area. Synthetic data constrain coordinates 1 and 2 of the structural frame, which is coherent with the data spatial distribution.**

To show the effects of the conceptual models in the results, we present two 3D geological models of the Voisey's Bay intrusion, each of them constrained with a different conceptual model of its floor geometry. Both models are constrained using the ellipsis equation as the lateral contact conceptual model $C_L^\bullet$, similar to the previous case study (section 4.1). Model A is constrained using the equation of an oblique cone as $C_V^\bullet$, while model B is constrained using a constant function.

$$C_V^{\bullet\text{Model A}} : c_0 = f(\varphi(c_1', c_2'), (c_0^v, c_1^v, c_2^v)$$
$$C_V^{\bullet\text{Model B}} : c_0 = c_0^v$$

Where $\varphi(c_1', c_2')$ is the conic guiding curve of the cone, and $(c_0^v, c_1^v, c_2^v)$ are the intrusion frame coordinates of the deepest data point, which in Model A acts as the vertex of the cone. Figure 6 shows the resulting 3D models.





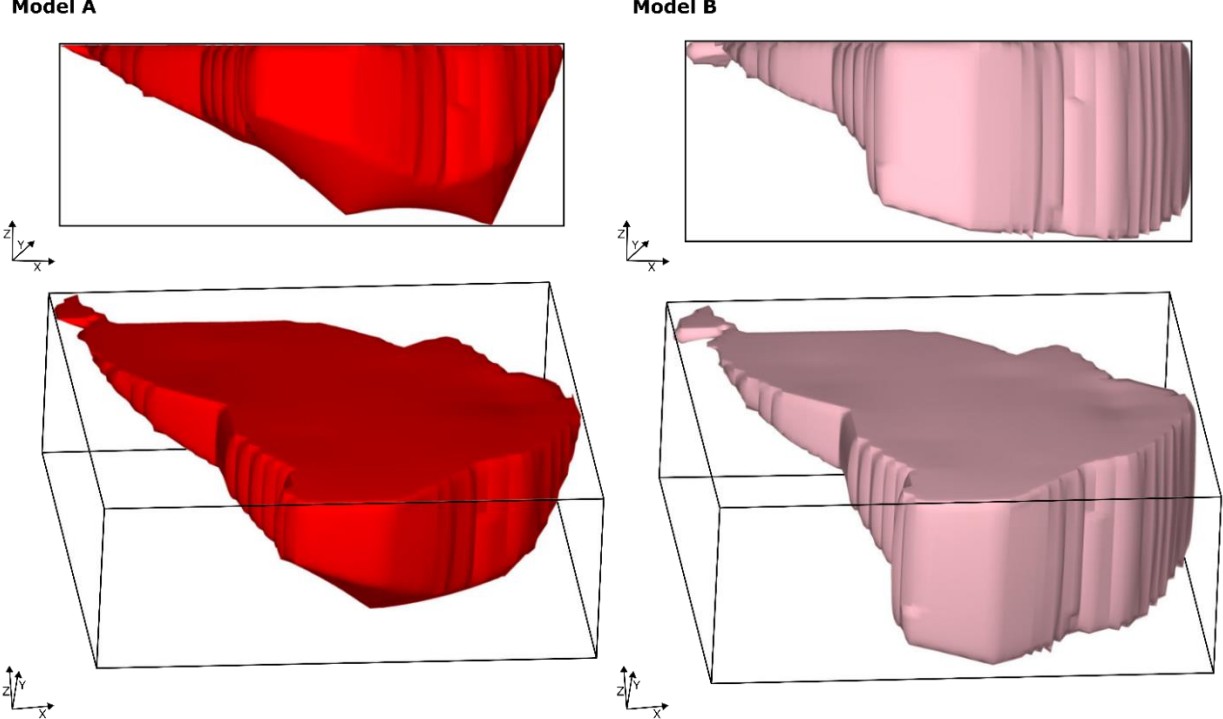

**Figure 6. 3D geological models of Case Study 2 – Voisey's Bay intrusion, Canada. Model A and model B are built using the same input data and the ellipsis function to constrain their lateral contact. The difference between them is the function that limits their vertical contact. Model A is constrained using the cone function, while model B is constrained using a constant function.**

### 4.3 Case Study 3: Synthetic laccolith

The third case study is a synthetic laccolith emplaced in a horizontal stratigraphic sequence. The input data consisted of an implicit geological model of the stratigraphy, 6 data points of the roof and floor contact of the intrusion, and a point in the middle of the intrusion that indicates the laccolith long axis' position and a propagation direction parallel to the long axis. The intrusion frame is built using the floor contact, propagation and long axis data, and its coordinate 0 is constrained to be parallel to the host rock bedding. Figure 7 shows the 3D geological model of the host rock, the distribution of the laccolith data, and the intrusion frame.





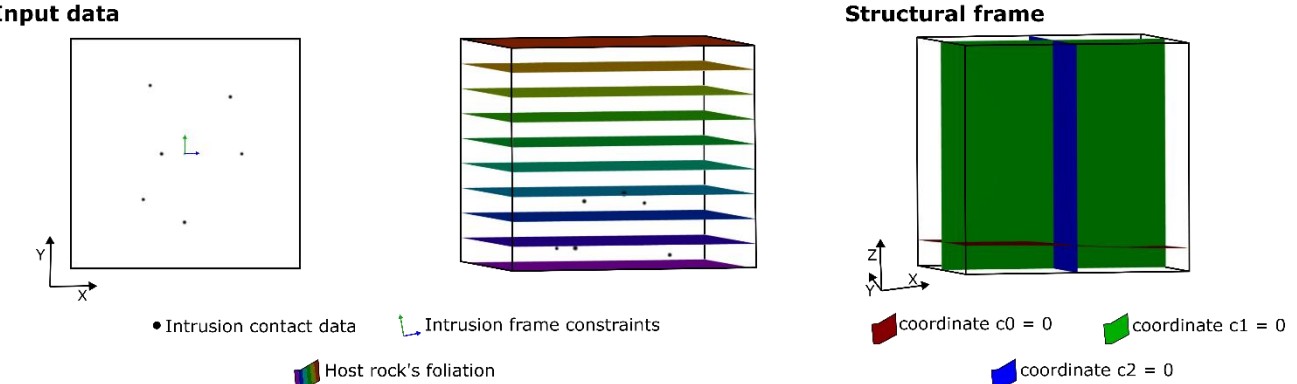

**Figure 7. Input data and structural frames of Case Study 3 – Synthetic laccolith. The dataset consists of the 3D model of the host rock, six roof and floor contact points, and one point and vector to constrain coordinates 1 and 2 of the structural frame.**

The conceptual models used in this example are the ellipsis equation as the lateral contact conceptual model $C_L^{\circ}$ (same as case studies 1 and 2) and a bell curve function as the vertical conceptual model $C_V^{\circ}$.

$$C_V^{\circ}: c_0 = \frac{1}{a\sqrt{2\pi}} * \exp(-\frac{1}{2}\frac{(c_1-b)^2}{a^2})$$

Where $a$ is the maximum half distance between the data points along $c_1$ and $b$ is the middle point along $c_1$, considering the spatial distribution of the data.

The threshold function $T_V$ characterises the thickness variation of the intrusion as distances along $c_0$ for any $(c_1, c_2)$. To reproduce the effects of the intrusion emplacement by roof lifting into the host rock, we use $T_V$ to modify the geometry of the

horizontal stratigraphy, so it is concordant to the intrusion roof. Figure 8 shows the resulting 3D model, and a cross-section of the model illustrating the geometry of the host rock after the emplacement of the intrusion.





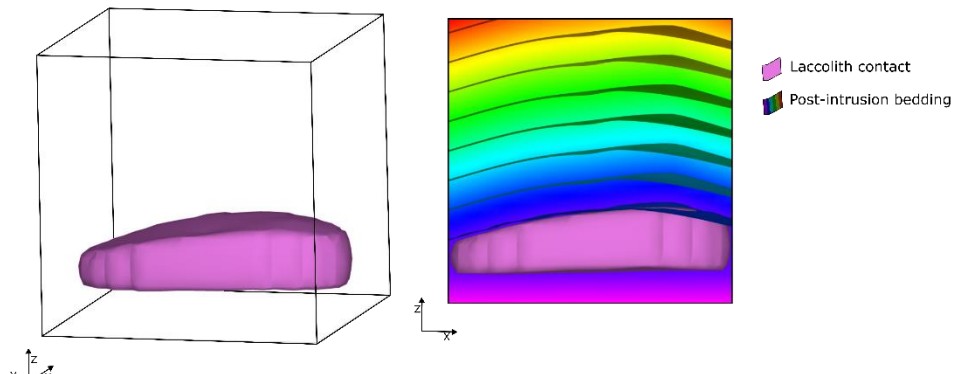

**Figure 8. 3D geological model of Case Study 3 - Synthetic laccolith. To the right, a cross-section shows the geometry of the bedding folded using the geometry of the laccolith roof.**


## 4.4 Comparison with Radial Basis Function interpolation

Radial Basis Function interpolation is one of the main approaches currently used to build implicit 3D geological models (Hillier *et al.*, 2014; Cowan *et al.*, 2002; Wellman and Caumon 2018). To assess the value of our approach, we present a comparison between our method and Radial Basic Functions interpolation. We apply both methods to build 3D geological models of a sill

intrusion in the offshore NW Australia shelf (Case Study 4, Köpping *et al.*, 2022). This real-world case study is an exceptional example to perform the comparison because it is extensively mapped in seismic images, and its geometry is well characterised. We built four 3D geological modes for this case study, whose differences arise from the method used to build them and the amount and type of input data. Models A and B are built using Radial Basic Functions interpolation and differ in the number of input constraints for each model. Models C and D are built using our proposed method, and the difference between them is

that model D incorporates geometrical constraints from the emplacement history proposed by Köpping *et al.* (2022).

The input contact data for the models is a randomly-selected sample of the dataset presented by Köpping *et al.* (2022). The original dataset consists of the sill base and top contact points picked from seismic imagery and covers approximately 4042 km$^2$ with > 2.5 million data points (Figure 9a). According to Köpping *et al.* (2021, Figure 9), the intrusion is composed of a

13.4 km long, N-trending and strata concordant inner sill, which transitions into transgressive inward-dipping inclined sheets along its eastern margin and southwestern margin. Where inclined sheets are developed, the horizontal dimension of the inner sill is relatively narrow (~3.4 km). In the north section of the sill, where no inclined sheet is developed on the western margin, the inner sill widens up to 6.4 km and has a convex-outwards and lobate western termination. The authors present a detailed characterisation of the vertical thickness variation within the sill (Figure 9b). The eastern half of the inner sill is ~166 to ~249

m thick, rapidly decreasing westward to ~111 to ~166 m. The inclined sheets, the southern sill tip and the northwestern lobate





termination, are presented as tuned reflection packages, and their thickness can only be defined by the limits of separability and visibility of the data (~7 to ~56 m).

Köpping *et al.* (2022) propose an emplacement model for the sill, schematically represented in Figure 9c. The sill comprises
one segment that propagated and inflated northward from a SW-NE trending fault and another segment that propagated to the southwest of this fault. This SW-NE trending structure is located in the middle of the sill and likely also facilitated magma ascent. The transgressive inward-dipping inclined sheets formed along pre-existing faults in the east and south-west. The straight geometry of the southwestern limb is interpreted to be controlled by pre-existing fractures and/or faults.

The pre-processing of the data, workflow and results of the four models are presented in the following subsections. The input data and resulting 3D models are presented in Figure 10.



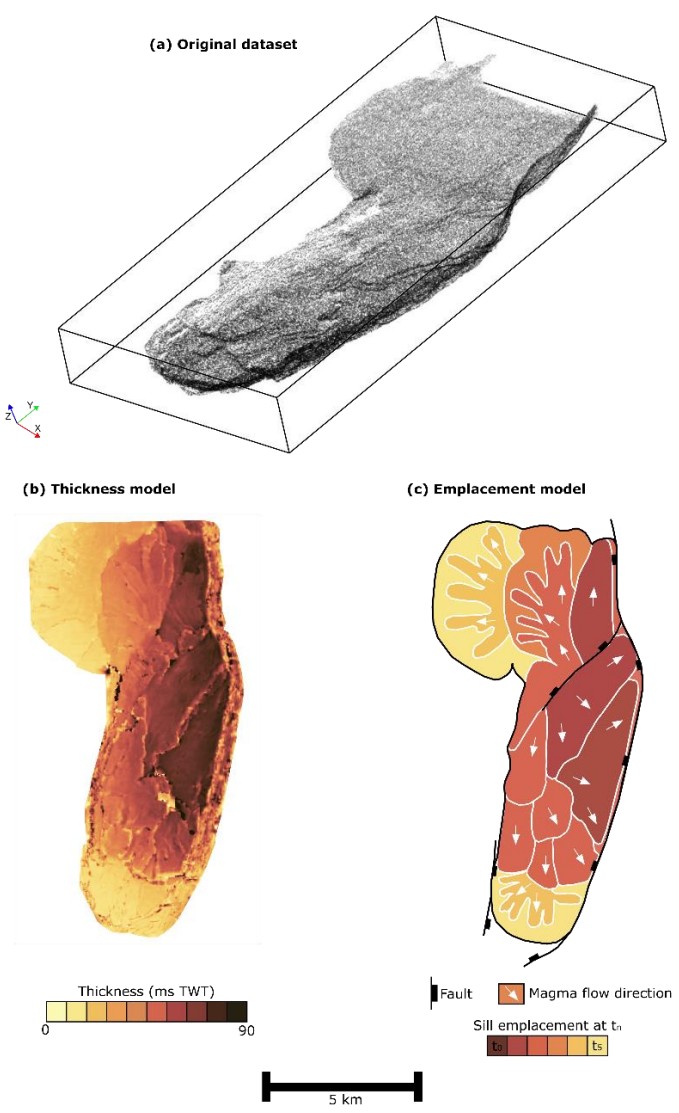

**Figure 9. Data and models of Köpping *et al.* (2022): (a) Top and base contacts points picked on seismic images, (b) two-way time thickness model, and (c) schematic diagram of the emplacement history of the sill.**





**Model A and B - Radial Basis Function (RBF) implicit interpolation**

Models A and B were built using the *SurfE* interpolator available in *LoopStructural* (Grose *et al.*, 2020, 2021a). *SurfE* (*https://github.com/MichaelHillier/surfe*) implements a generalised radial basis function interpolator (Hillier *et al.*, 2014). Radial basis function interpolation is a meshless interpolation and the scalar field can be constrained with different types of data, including value and gradient constraints. Models A and B are built using the signed distance interpolation of *SurfE* (single surface method).

The input data for these models consist of value and gradient constraints. In both models, the value constraints represent the intrusion contact location, and a value of 0 is assigned to each of these points. Gradient constraints correspond to vectors perpendicular to the stratigraphy with a direction towards the outside of the intrusion. For model A (Figure 10), a sub-sample of approximately 0.1% of the original dataset is used as value constraints, and a selection of these points located in the strata concordant inner sill is used as gradient constraints. For model B (Figure 10), we increase the amount of value and gradient constraints to approximately 0.5% of the original dataset. In particular, the gradient constraints are distributed within the inner and outer sills.

**Model C and D - Structural frame and conceptual models**

Models C and D are built using the approach introduced in this work. The main difference between these two models is that model D integrates geometrical constraints from the sill emplacement history proposed by Köpping *et al.* (2022). In other words, we use the geometry of the faults that facilitated the emplacement of the transgressive sills and the conceptual propagation model proposed by Köpping *et al.* (2022). The resulting 3D models are shown in Figure 10.

The contact data for both models consist of a sample of approximately 0.1% of the original dataset, the same data points used for model A. These points are classified as top, base and lateral contacts depending on their location. The intrusion frame $c_0$ is built using the sill's base contact points and is constrained to be perpendicular to the host rock. To constraint coordinates c1 and $c_2$, we approximate the long axis of the intrusion considering the spatial distribution of the data. The gradient of $c_1$ and $c_2$ are constrained to be parallel and perpendicular to the long axis, respectively (Figure 10).

For model D, we consider the sill composed of two segments emplaced at opposite sides of a NE-SW striking fault (Köpping *et al.*, 2022, Figure 10). The northern segment propagates into the fault's hanging wall towards the N-NW, and its geometry is controlled by the eastern marginal fault generating a transgressive sill. The southern segment propagates within the footwall towards the SE and then SSW. The transgressive sills to the east and west of the southern segment are controlled by pre-existing faults. The two segments are modelled separately. For both segments, the intrusion frame $c_0$ is built using the sills' base contact points and is constrained to be parallel to the host rock and the marginal faults involved in their emplacement.





The propagation vectors given by the emplacement model of Köpping *et al.* (2022) are used to constrain $c_1$, and $c_2$ is constrained using a point located in the middle of the sill and a gradient perpendicular to the propagation direction at that location (Figure 10).


Models C and D are built with the same conceptual models: the ellipsis equation as the lateral contact conceptual model $\overset{.}{C_L}$ and a constant function as the vertical conceptual model $\overset{.}{C_V}$.

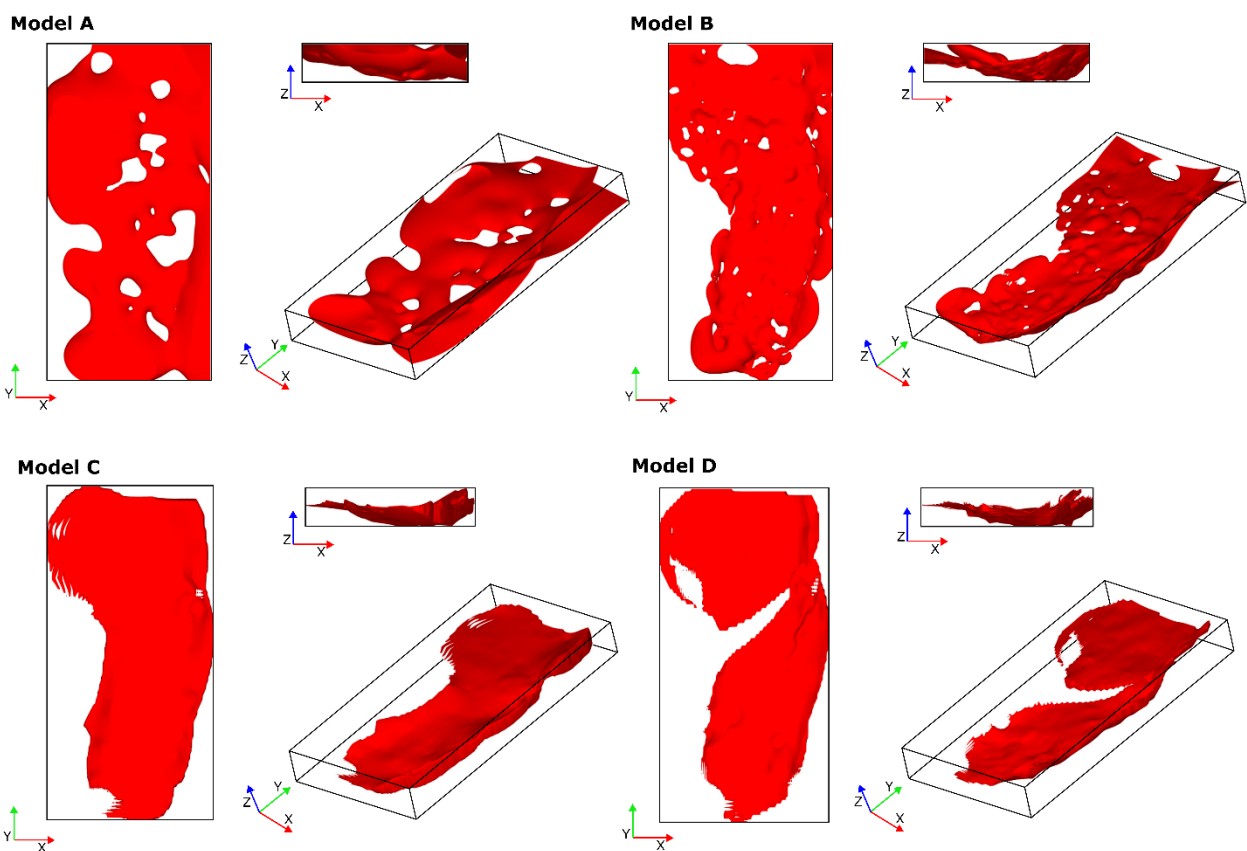

**Figure 10. 3D geological models of Case Study 4 – Sill complex in Western Australia. Models A and B were built using Radial Basis Functions interpolation, and Models C and D were built using the method proposed in this work.**

**Comparison between the models**

Visual inspection shows that, in general, RBF interpolation and our method can reproduce the coarse-scale geometry of the sill, with a N-trending inner sill transitioning to inward dipping outer sills. Considering the geometric description provided by

Köpping *et al.* (2022, Figure 9), our method is more accurate at constraining the shape of the terminations of the sill, while the





RBF interpolation extrapolates the isosurface that represents the intrusion contact away from the data. This is exacerbated in Model A due to the reduced number of input data compared to model B. In RBF interpolation, the value of the basis function depends on the distance $\|x - x_i\|$ where x is the position to evaluate the function and $x_i$ the location of the data point, which may generate blobby geometries away from the data (Wellmann and Caumon, 2018). Models A and B present holes within

the intrusion related to the absence of on-contact or planar constraints. Models C and D do not capture some of the sill thinnest parts, such as the southern tip, the northwestern lobate termination and the eastern inclined sheet of the northern segment of model D. In these areas, the grid elements have larger dimensions than the width or length of the sill, and therefore the isosurface representing the intrusion contact is not captured in the scalar field values assigned to each grid nodes.

To assess how realistic the resulting 3D models are, we compare the geometry given by the seismic imagery and the geometry given by each model. We visually inspect 12 cross-sections and measure each model's thickness. As an example, Figure 11 shows one cross sections along the X-axis and one cross section along the Y-axis. Model A shows substantial differences compared to the other models, and it does not reproduce the expected sheet-like shape of a sill nor a clear transition from the inner to the outer sill. Models B, C and D capture the inclined geometry of the outer sills; however, model C seems to flatten

the eastern inclined sheet. This is because model C's intrusion frame is interpolated using the base contact points, and this interpolation does not necessarily capture the geometry of the faults that control the transgressive sill. Models C and D are slightly better at recovering the straight top and base contacts, while model B exhibits wavy contacts in some parts of the model.

The thickness of the models is measured in pre-defined locations and compared with the thickness given by the seismic imagery observations. Figure 12 shows the location of the measurements and thickness contours interpolated using these measurements of each of the models. As Köpping *et al.* (2022) describe, their data shows that the intrusion thickness decreases from E to W within the inner sill and towards the tips and inclined outer sills. Model A does not show any evident trend, and the thickness is generally larger than the thickness given by the data. Model B thins down towards the western lobate termination but does

not capture the decreased thickness observed in the outer sills and the southern tip. Models C and D show a decreasing trend towards the western and southern tips but tend to amplify the difference with the data closer to the outer sills. We compute the difference between the thicknesses measured on each model and the thickness given by the data (Figure 12, Table 1). This difference's mean and standard deviation are significantly lower in model B with respect to model A, showing the effect of adding more constraints to the RBF interpolation. Model C and D have a similar mean and standard deviation, and these figures

are slightly lower in model C.



**Table 1. Input data and results of the thickness comparison between the models of Case Study 4.**

|  | | Input data | | Thickness difference | |
|---|---|---|---|---|---|
|  | N° of on-contact constraints | N° of planar constraints | N° of intrusion frame constraints | Average | Standard deviation |
| **Model A** | 184 points | 88 vector data | 0 vector data | 459.4 | 409.9 |
| **Model B** | 755 points | 570 vector data | 0 vector data | 108.2 | 167.0 |
| **Model C** | 184 points | 0 vector data | 22 vector data | 61.2 | 124.8 |
| **Model D** | 184 points | 0 vector data | 27 vector data | 32.8 | 56.2 |


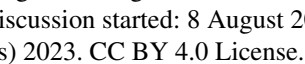



**Figure 11. Assessment of the 3D geological models of Case Study 4 – Sill complex in Western Australia. The figure shows cross-sections of the models along 192675E and 7786589N. Models A and B were built using Radial Basis Functions interpolation, and models C and D were built using the method proposed in this work. The first row of polygons (Data) shows the area enclosed by the dataset presented by Köpping *et al.* (2021, Figure 9).**






**Figure 12. Assessment of the 3D geological models of Case Study 4 – Sill complex in Western Australia. The figure compares the models' thicknesses (first row) and the difference between the thickness given by the data and the thickness given by the models (second row). Models A and B were built using Radial Basis Functions interpolation, and models C and D were built using the method proposed in this work. The figure to the left (Data) shows the location of the thickness measurements and thickness contours estimated using thickness data of Köpping *et al.* (2022). The other figures show thickness and thickness-difference contours estimated using the measurements on each model. The estimated contours are clipped using the outline of each model.**





## 6 Discussion

To date, 3D models of intrusions are built with classical interpolation workflows, where on-contact data and a polarity constraint indicating the inside/outside of the intrusion are used to estimate the contact. Post-processing is usually required to generate the intrusion shapes observed in the field, drilling data, or imaged in geophysical surveys, making the model dependent on the modeller's expertise and challenging to reproduce. In this contribution, we address these limitations by implementing a method inspired by the Object-Distance Simulation Method (Henrion *et al.*, 2008; 2010) and that uses an adapted structural frame for intrusions (Laurent *et al.*, 2013, 2016; Godefroy *et al.*, 2017; Grose *et al.*, 2021a, b). The models can be constrained with contact data and other field measurements such as inflations direction and propagation direction.

The structural frame incorporates conceptual knowledge of intrusion emplacement mechanisms into the modelling framework. This is achieved by constraining the structural frame with the geometry of the foliation or geological structures that facilitated the emplacement of the intrusion. Thus, the geometry of the modelled intrusions is controlled by the geometry of the host rock. It may also be constrained with the inflation and propagation direction, if this data or conceptual knowledge is available. The intrusion frame allows characterising the geometry of intrusions more simply. For example, a saucer-shaped (e.g., CS4) sill becomes a straight, tablet-shaped sill viewed along the coordinates of the intrusion frame. This is particularly useful for complex systems of intrusions, such as sills that step up and down within the stratigraphy with variable propagation directions. Figure 13 compares the thickness and width variation between the magma lobes of the CS1 (Section 3.1). This case study illustrates a synthetic sill complex comprised of three magma lobes propagating in slightly different directions. The middle sill steps up, exploiting a pre-existing structure. The plots in Figure 14 show how the thickness and width of each sill vary while they propagate away from the feeder.





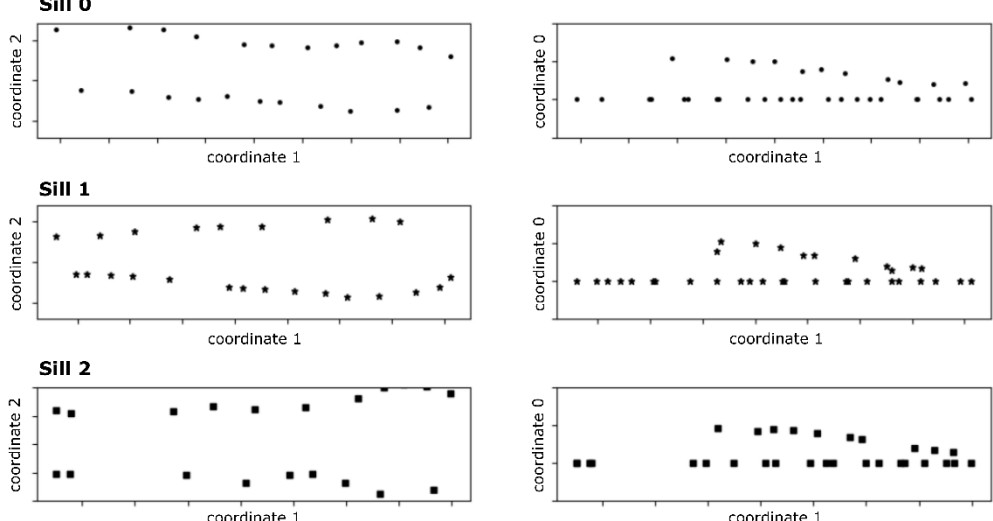

**Figure 13. Visualisation of contact data along the structural frame axes.** The figure shows the contact data from Case Study 1 (synthetic sill complex) which is plotted along the frame coordinates. Coordinate 1 plotted against coordinate 2 (left) shows the width variation for each sill. Coordinate 1 plotted against coordinate 0 (right) shows the thickness variation for each sill.

The intrusion frame coordinates are employed to parameterise conceptual models that represent the coarse-scale geometry of the intrusion. The conceptual models represent a parametric description of the intrusion thickness and width, and the modeller defines these functions. They would be comparable to defining a conceptual model while drawing shapes or adding arbitrary (non-quantified through proper geostatistical analysis) structural trends to the model, but with no manual processing. Thus, the conceptual models allow building objective and unbiased 3D models considering prior knowledge of the intrusion geometry. While the method workflow accepts any parametric function, it is recommended that these functions agree with the geometries observed in reality (see Section 2). The appropriate function can be selected after assessing the data and the regional context. The conceptual models also allow testing different scenarios. In CS2, we create two models of the Voisey's Bay intrusion whose difference lies in the conceptual model. In one of them, we model a scenario where the intrusion has a wedge-shaped geometry using the function of an oblique cone to constrain the intrusion floor geometry. In the second model, we test a tablet-shaped geometry using a constant function to constrain the floor geometry. Both models comprise two alternatives for the geometry of the intrusion considering the spatial distribution of the data. A workflow to automatically find the best fitting conceptual model can be implemented in the future. Following the approach of Grose *et al.* (2018; 2019), fitting the conceptual model to the observations can be considered as an inverse problem. Finally, the conceptual models allow to build intrusion models with small contact datasets and in the absence of lateral data, as shown in case study 3. The use of off-contact data (i.e., inside or outside the intrusion) to constrain the fitted conceptual model can be considered for further implementations.



The structural frame together with the conceptual model allow us to have an implicit representation of the intrusion thickness
and width within the intrusion extent. This implicit representation can be used to modify the host rock to recreate the effects
of the intrusion emplacement in the host rock geometry. This is demonstrated in case study 3, where we modify the originally
flat-lying host rock to obtain a folded bedding concordant to the bell-shaped geometry of the laccolith roof. Further work
should consider demonstrating this capability with real-world case studies.

In general, the 3D models of the four case studies presented in this work are in good agreement with intrusions geometries
described in the literature (e.g.,Cruden *et al.*, 2017, 1999; Galland *et al.*, 2018; Jackson *et al.*, 2013; Kavanagh 2018; McCaffrey
and Petford 1997; Vigneresse 1995; Vigneresse *et al.*, 1999). These examples demonstrate the capability of our method to
reproduce intrusions geometries, in particular, the coarse scale geometry of sill segments and the connectors developed after
the interaction between sill segments in a sill complex, the coarse-scale geometry of plutons, with a generally flat roof with a
symmetric or asymmetric floor and the bell-shaped geometry of laccolith. The examples also show that the method can create
realistic intrusion shapes considering small datasets from surface or drilling data. The modelling workflow for other intrusion
types, such as dykes or lopoliths, would be similar, with the main difference being the conceptual model defined for each case.

Considering our Case Study 4 (Section 5), our method can reproduce more truthfully the sill geometries imaged in seismic
surveys, compared to RBF interpolation. In particular, our method can replicate the sheet-like geometry of this sill intrusion,
constrain its terminations and thickness variations, and generate a model of similar dimension, including thickness variation
trends, to what is observed in contact data. Parameterisation of the intrusion using the structural frame is crucial and enables a
rigorous computation of the intrusion extent in the direction in which the intrusion grew. In our approach, it is also possible
for the modeller to add geometrical constraints knowing the emplacement history of the intrusion. For this case study, in Model
D we were able to model the steeply inclined sheets by constraining the intrusion frame to be parallel to the marginal faults
that facilitated the emplacement of the transgressive sills. This type of geometry would be difficult to reproduce using a classic
interpolation approach unless a large dataset was provided, as in Model B. However, having a dense dataset is rarely the case,
and models of intrusions are usually built using sparse and unevenly distributed datasets. The models built using RBF
interpolation may be improved by modifying the distance scalar field with an elliptical conceptual model. Nevertheless, this
is out of the scope of this work.

The computing time of adding an intrusion to the 3D models ranges from 3 to 20 seconds. The computing time is proportional
to the size of the grid, and the number of geological features (e.g., bedding, faults) used to constraint the intrusion frame. The
computing time of building the 3D geological models presented in this work, including their visualisation, ranges from 15
seconds to 3 minutes. All the models were built in a consumer laptop PC.



The main limitation of the proposed method is that the surface representing the intrusion contact depends on the size of the model grid elements. Consider a part of the intrusion that is narrower or thinner than the size of a grid element, in this case, the nodes around the intrusion will indicate threshold values $T_V$ and $T_L$ smaller than their respective $c_2$ and $c_0$ coordinates, and they will not be indicated as being inside the intrusion. The scalar field value on these nodes will be greater than 0, and therefore no isosurface 0 will be found between them. This scenario is observed in the narrower zone of Voisey's Bay intrusion model (CS2, Figure 8). According to the data, the intrusion transitions to a narrow and thin sill-like intrusion, which the model does not capture. This is also observed in the thinnest parts of the sill intrusion in NW Australia presented in section 5 (CS4, models C and D in Figure 10). This limitation can be addressed using a higher resolution mesh, however this introduces computing limitations (time and memory usage). Adaptive meshing algorithms should also be considered in the next iteration of the implementation.

**7 Conclusions**

Current methods to build 3D models of igneous intrusions are strongly dependent on data availability and manual processing. They do not consider geological knowledge of intrusion emplacement mechanisms objectively and do not use all types of measurements collected in the field. In this context, the generation of intrusion shapes observed in the field and in geophysics imagery is challenging to reproduce. To address these problems, we developed a method to build 3D models of intrusions that accounts for geological knowledge on intrusion emplacement mechanisms and typical datasets. The method is inspired by the Object-Distance Simulation Method (ODSIM) and incorporates an intrusion structural frame into the ODSIM framework that accounts for intrusion growth and propagation. This structural frame provides a curvilinear coordinate system for each intrusion within the model. Conceptual models of the intrusion contacts are parameterised using the structural frame coordinates and then fitted to the data. The conceptual models include a conceptual idea of the intrusion shape objectively and allow to test of different scenarios without the modeller's bias. The intrusion and the conceptual model provide a characterisation of the intrusion thickness and width that may be used to alter the host rock to 3D model the deformation associated with the intrusion emplacement. Fitting of all the data is not always feasible and may be dependent on the grid size. Further work on the method will include automatically fitting the conceptual models to the data, incorporating off-contact data, and employing adaptive meshes to improve the intrusion resolution.

**Code and data availability**

The examples presented in this contribution were generated using the open source 3D modelling package LoopStructural. LoopStructural v.1.5.10 can be downloaded from *https://zenodo.org/record/7734926* or installed using *pip install LoopStructural*. The input data and Jupyter notebooks of all the examples presented in this work can be downloaded from *https://doi.org/10.5281/zenodo.8189191*.





**Author contributions**

All authors contributed to the conceptual design of the method and comparative analyses presented in this work. FAN
developed the model code with editing and improvements contributions from LG. FAN prepared the manuscript with editing
and reviewing contributions from all co-authors.

**Competing interests**

The contact author has declared that none of the authors has any competing interests.

**Acknowledgements**

The authors thank Italo Goncalves and an anonymous reviewer for their insightful comments. Funding was provided by ARC
Linkage grant LP17010985 – Enabling 3D Stochastic Geological Modelling; supporting the development of the Loop platform.
Ms Alvarado is funded by Monash International Tuition Scholarship. The authors would like to also thank Jonas Köpping for
providing the data of case study 4.

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
