# Peer review of "3D geological modelling of igneous intrusions in LoopStructural v1.5.10"

_Geoscientific Model Development, 2023_

## Referee Comment (RC3)

[referee-annotated manuscript omitted]

---

## Author Response (AR1)

Dear referees,

Thank you very much for thoroughly reviewing the manuscript and for your recommendations. Please, see below our response to your comments and a list of the main changes in the manuscript.

**1.1. Comments from Referee 1 – Anonymous Referee**

Referee comment number 1 was responded through the interactive discussion portal. Thus, we respond here to their comment number 2, which refers to the interpolation technique employed in the data conditioning step of our method

*"SGS is an exact interpolator, but to make that work in this case you need to meet the conditions of that interpolator, which are similar to the kriging interpolator. I think the issue of interpolation is not covered very clearly, at least, it was difficult to understand what exactly was done, so I suggest that you make this more clear in the next version, likely discuss the limitations, or show with an example what the limitations are."*

**1.2. Response to Referee 1 – Anonymous Referee**

As stated in the interactive discussion, our method does not employ Sequential Gaussian Simulation (SGS). The method employs an exact interpolation technique such as Radia Basis Function, which differs from Kriging since it does not require the definition of a variogram model. However, we agree that the description of the data conditioning was not detailed enough, and we have improved the description of this step of the method in section *3.3. Conceptual models and threshold functions*. Now, the manuscript described in three sub-steps how the conceptual models are conditioned to the data and how the threshold functions.

The main limitation of RBF and other exact interpolations are their restricted capacity of handling large datasets of thousands of samples (e.g. Calcagno et al., 2008). However, this work has been developed specifically considering datasets consisting in field measurements or other punctual data, which are not usually large datasets. Therefore, this limitation is not considered as a limitation of the method.

**2.1. Comments from Referee 2 – Gautier Laurent**

*"A couple of particular comments even if quite minors:*

1. *there seem to be an issue in equationp10 line 220, probably missing the absolute value symbol?*
2. *in section 4.3, p15 line 320: the process to edit the stratigraphy to account for the intrusive body shape is not clearly described, but would be very interesting*
3. *there is a typo in the equation line 288 p 13*
4. *the colormap for the thickness difference in figure 12 is a bit misleading and would be diffcult for colorblind peoples"*

**2.2. Response to Referee 2 – Gautier Laurent**

1. Equation was fixed, and absolute symbol was added.
2. We have added a brief description on how the pre-intrusion stratigraphy was modified using the geoemtry of the modelled laccolith (section 4.3 Case Study 3: Synthetic laccolith)
3. Typo was ammended.
4. We have modified the colour map of Figure 12 to a colourblind-friendly colour map (red to blue colourmap).

**3. Other minor changes in the manuscript.**

This manuscript is part of F. Alvarado-Neves PhD thesis, which went under peer-review late last year. The following changes are after this review, so the manuscript for GMD is consistent with the manuscript in the thesis.
- Minor changes in figures, such as replacing XYZ arrow for North arrow in real-world examples or where necessary, legend added where missing.
- Section 2: clarification that intrusion emplacement is not always controlled by host rock anisotropies. This is also added in the Section 5 – Discussion, and its impact in the method.
- Section 3: Minor changes in the method overview and structural frames description, to improve the clarity of the method's description.
- Section 4.4: The specifications of the grid employed for the models visualisation and a figure of the input data distribution were added. A discussion on the difference between models and estimated thickness was added (Page 23, L459).
- Section 5: We have added a discussion on (i) how the method could be applicable to intrusions whose emplacement was not controlled by mechanical anisotropies, (ii) adding conceptual knowledge through the conceptual models, and (iii) the fact that the method does not use off-contact data.
- Update of links to Zenodo repository. Jupyter notebook have now available the option to run them in GoogleColab.